# A Custom-Made Lower Limb Dynamometer for Assessing Ankle Joint Torque in Humans: Calibration and Measurement Procedures

**DOI:** 10.3390/s22010135

**Published:** 2021-12-25

**Authors:** Iulia Iovanca Drăgoi, Florina Georgeta Popescu, Teodor Petrița, Romulus Fabian Tatu, Cosmina Ioana Bondor, Carmen Tatu, Frank L. Bowling, Neil D. Reeves, Mihai Ionac

**Affiliations:** 1Department of Vascular Surgery and Reconstructive Microsurgery, “Victor Babeș” University of Medicine and Pharmacy, 2 Eftimie Murgu Square, 300041 Timisoara, Romania; masteriuliadragoi@icloud.com (I.I.D.); frank.bowling@manchester.ac.uk (F.L.B.); mihai.ionac@umft.ro (M.I.); 2Discipline of Occupational Health, “Victor Babeş” University of Medicine and Pharmacy, 2 Eftimie Murgu Square, 300041 Timisoara, Romania; 3Department of Communications, Politehnica University Timișoara, 2 Vasile Pârvan, 300223 Timisoara, Romania; 4Department of Orthopaedics-Traumatology-Urology and Imagistics, “Victor Babeș” University of Medicine and Pharmacy, 2 Eftimie Murgu Square, 300041 Timisoara, Romania; tatu.fabian@umft.ro; 5Department of Medical Informatics and Biostatistics, University of Medicine and Pharmacy “Iuliu Hațieganu”, 8 Victor Babeș, 400000 Cluj-Napoca, Romania; cbondor@umfcluj.ro; 6Department of Functional Sciences, Physiology, Center of Immuno-Physiology and Biotechnologies, “Victor Babeş” University of Medicine and Pharmacy, 2 Eftimie Murgu Square, 300041 Timisoara, Romania; carmen.tatu@umft.ro; 7Department of Surgery & Translational Medicine, Faculty of Medical and Human Sciences, University of Manchester, Oxford Rd., Manchester M13 9PL, UK; 8Research Centre for Musculoskeletal Science & Sports Medicine, Department of Life Sciences, Faculty of Science and Engineering, Manchester Metropolitan University, Oxford Rd., Manchester M1 5GD, UK; n.reeves@mmu.ac.uk

**Keywords:** ankle torque, dynamometer, muscle strength, calibration, linear regression, reliability

## Abstract

Custom-made dynamometry was shown to objectively analyze human muscle strength around the ankle joint with accuracy, easy portability and low costs. This paper describes the full method of calibration and measurement setup and the measurement procedure when capturing ankle torque for establishing reliability of a portable custom-built electronic dynamometer. After considering the load cell offset voltage, the pivotal position was determined, and calibration with loads followed. Linear regression was used for calculating the proportionality constant between torque and measured voltage. Digital means were used for data collection and processing. Four healthy consenting participants were enrolled in the study. Three consecutive maximum voluntary isometric contractions of five seconds each were registered for both feet during plantar flexion/dorsiflexion, and ankle torque was then calculated for three ankle inclinations. A calibration procedure resulted, comprising determination of the pivotal axis and pedal constant. Using the obtained data, a measurement procedure was proposed. Obtained contraction time graphs led to easier filtering of the results. When calculating the interclass correlation, the portable apparatus demonstrated to be reliable when measuring ankle torque. When a custom-made dynamometer was used for capturing ankle torque, accuracy of the method was assured by a rigorous calibration and measurement protocol elaboration.

## 1. Introduction

Normal gait and gait-related disorders have always been of great interest for the research community. Muscle forces acting around the ankle joint have been extensively studied to better understand human foot biomechanics and locomotion. Ankle range of motion (ROM), muscle parameters and the relation between ROM and torque around the ankle joint could help in explaining normal foot function and enable a better understanding of pathological gait. The plantar flexor muscle group acting around the ankle joint is the main participant in the contact phase during walking [1]. Triceps surae, with the gastrocnemius muscle particularly considered the main plantar flexor, and the toes flexor muscles (TFM) produce internal plantar flexion moment around the ankle joint. Flexor hallucis longus, flexor digitorum longus and the long toe flexor muscles also function to plantarflex the foot, and measuring the activity and strength of this particular muscle group is of relevance for understanding walking performance. Strength around the ankle joint has previously been measured using manual muscle testing (MMT), and for quantitative analyses, handheld dynamometry, electronic dynamometry or isokinetic testing have been used. 

Foot strength manual muscle testing has shown low reliability [2], while isokinetic testing is considered the gold standard due to its high reliability, however, it involves higher costs [3]. 

Plantar flexor strength and ankle torque can reach very high values [4], which poses a challenge for clinicians in maintaining the sensitivity of MMT and handheld dynamometry when testing these particular muscle groups. For this purpose, accurate and economic clinician-independent measurements for assessing foot muscle strength are needed. In striving towards greater precision while still maintaining low costs, different devices have been developed, and for both research and clinical practice, custom-made dynamometers have taken a more prominent role in foot and ankle strength evaluation. 

Few papers have assessed foot muscle strength in healthy adult participants, and normative quantitative data on isometric muscle strength were published [5]. 

In clinical rehabilitation practice, when isokinetic testing is not available, electronic dynamometry can be a valuable way to accurately measure foot strength for diagnostics, rehabilitation progression and treatment outcomes. Ankle torque can be accurately assessed in bilateral symmetrical conditions affecting gait, such as diabetic peripheral neuropathy, neurological degenerative diseases and systemic inflammatory conditions. Unilateral situations derived from trauma, orthopedic surgery events, sport injuries and focal neuropathies could benefit from a precise measurement, having the unaffected side as a control. Analyzing torque at different ankle angles could help establish the weakest ranges and further prescribe more specific exercise programs [6].

The first reference of a device which assessed the influence of joint position on ankle dorsiflexion in humans used a foot plate on pivots to measure ankle dorsiflexors’ strength and was described by Marsh in 1981 [6]. Another device with a force cell on a foot plate measuring forces acting around the ankle joint was a custom-build dynamometer similar to the one used by Marsh [7]. A device comparable to the one used by Reeves et al. in 2005 [7] was used to measure ankle torque during dorsiflexion and plantar flexion in healthy participants [8]. A fixed dynamometer was developed to determine external moments around the metatarsal phalangeal joints (MPJ) and was built in a manner similar to that of the ankle dynamometers mentioned previously [9]. Although some of these devices seem to use similar principles, specific data are not reported by the authors about the voltage measurement solution used for the calibration procedure. Moreover, the published papers present single data points rather than time graphs. Some of the papers use alternative current (AC)-provoked contractions instead of MVIC. All isometric dynamometers use a footplate with one or two load cells. Moraux [8] seems to use a similar signal processing chain including a low-pass filter, although no time recordings are published. Goldmann [9] used a PC digitizer card (NI 6024E, 12-bit ADC—manufactured by National Instruments, Austin, TX, USA.), without published time graphs. Marsh [6] briefly accounted for an unspecified type oscilloscope usage, but showed only singular contraction time graphs.

As there is no widely accepted method for measuring foot and ankle strength [10], improving the already existing methods could enhance muscle testing procedures. Little data were detailed in previous published papers on the calibration procedure of the used electronic custom-made dynamometers and the determination of the pivotal point as the axis of rotation for measuring ankle torque in humans [6].

The apparatus used in the present study is a replica of the device used by Reeves et al. [7], and a calibration procedure for the given device was recommended by the manufacturer before trials. 

Establishing the calibration procedure for the existing portable custom-made electronic dynamometer and standardization of a method of measurement of strength around the ankle joint were the main objectives of this study.

Another objective was to determine ankle torque during maximal voluntary isometric contractions (MVIC) for both plantar flexors and dorsiflexors muscle groups and establish reliability for this portable dynamometer.

The novel contribution of our study is a new method of calibration for portable dynamometry devices using weights of known mass to calculate the proportionality constant through linear regression to precisely establish the apparatus’ pivotal point (central of axis of rotation). The innovation in this study was introducing torque-time graphs for the resulted torque measurements by mathematical data processing of the oscillograms. The experimental setup in the present study—with the oscilloscope software having the advantage of producing torque-time graphs and recording the whole movement. We found oscillograms very important since they allowed for detection of failed measurements due to unintentional movements of the participants (e.g., incorrectly directed plantar/dorsiflexion or incorrectly timed contractions). Torque-time contraction graphs open possibilities for more in-depth analysis.

## 2. Materials and Methods

### 2.1. Description of the Device

An aluminum pedal was suspended on a shaft at one end and on a weight-measuring load cell close at the other end. While the pressure was pointed to the load cell, the pedal had no possibility to move while fixed in position. The applied force on the load cell was assessed electronically. The pedal had the possibility to be pointed at various inclinations to evaluate ankle torque at different ankle joint angles. The pedal through its construction converted the torque to force applied on the load cell which was further converted into voltage. The pedal was built in such a manner to permit participant evaluation resting in a seated position. The knee could be fixed with a regular fixing strap, made from synthetic nonelastic fabric. Additionally, the foot could be fixed to the pedal in a similar manner. Generated muscle compression or decompression was read by the load cell. 

Our device used a common load cell CZL-601 (Figure 1) as the weight sensor with a strain gauge rated at 100 kg. The strain gauge could be connected as a classical Wheatstone (resistive) bridge. The load cell is available from a series of producers, ours being particularly this one [11], as chosen by the manufacturer. A single point load cell is a metal bar with strain gauges placed on it in such a manner that they can evaluate tension during muscle contraction. The dynamometer pedal as well as the load cell amplifier were hand-manufactured by Research Solutions Ltd., Alsager, UK [12]. The plate inclination was easily changeable so that torque could be measured at different ankle angles, the exact angle of dorsiflexion and plantarflexion being possibly set using an electronic inclinometer placed on the plate.

### 2.2. Description of the Measurement System 

The system consisted of four components, as shown in Figure 2 (right): a pedal with a load cell, load cell amplifier, universal serial bus (USB)-connected oscilloscope and a personal computer (PC) running the oscilloscope software.

The load cell was connected in resistive bridge configuration, hence it went through a four-wire cable to the load cell amplifier. The load cell amplifier converted the Wheatstone bridge imbalance to voltage, which was then evaluated with the oscilloscope. The oscilloscope connected to the PC and the oscilloscope software used in our case had the advantage of producing graphs of the recorded torque and memorizing the whole movement.

The oscilloscope used by the device manufacturer was a PicoScope model 2204A (manufactured by Pico Technology, St Neots, UK), with its software installed on our PC (we used a laptop PC). In our case, with the PicoScope2204A, we used PicoScope^®^6 software, freely available on the same producer’s website [13]. The alternative from the same producer is PicoLog^®^6 software, which functions as a data logger, but our experiments showed that it was not suitable for our purpose. PicoScope^®^6 software is more complex and allows particular parameter selection (e.g., frequency of acquisition, time of acquisition, resolution enhancement). The choice of configuration parameters is detailed in this paper.

Force was then evaluated by measuring the unbalance of the resistive bridge by electronic means. Calibration was needed prior to force measurement. In this paper, we disclose the full description of the calibration and the measurement procedure.

### 2.3. Calibration Procedure

In the beginning, distance marks were set on the pedal. Our pedal was fitted with a wood board with centimeter-marked ruler stickers on it (Figure 2, left). The distance markers were needed to set the pivotal point and then to position the calibration weights. Two dumbbells of known mass were used in our case. 

First, one needed to find out the pivotal point position and set a marker on the pedal; this would be the ankle joint center of axis of rotation determined by the medial and lateral malleoli. After this, one needed to measure the baseline offset voltage, which was the voltage measured by the system in absence of any foot load/pressure. The next step is making multiple measurements at different positions with different weights and after that performing linear regression with the results. The two dumbbells and the combination of them (the 0.5 kg dumbbell on the top of the 1 kg dumbbell) would be placed in various positions around the wood board, from the tip of the board until proximity to the pivotal position. It was not mandatory that the displacement step be constant, nor the registration of the offset voltage, due to the fact that the offset of the load cell would be calculated later by linear regression. The step should be 1.5–2.5 times the dumbbell edge.

The outcome of the linear regression was the pedal constant, which transformed the measured voltage into torque. There may have been some noise in the collected results (especially electric network noise) which could be discarded through low-pass filtering and other signal processing-specific means.

For measurement processing, we saved the results from the oscilloscope software and used GNU Octave software suite [14] which is regarded as a free alternative to Matlab [15]. We used Matlab for the regression and Octave for the other graphs. Our script was compatible with both suites.

### 2.4. Participant Characteristics

Four healthy volunteers gave written consent to participate in the study. 

Ethical approval from University of Medicine and Pharmacy “Victor Babeş” Timişoara Ethics Committee was released and registered under Nr. 50/21.09-14.10.2020.

Any past unilateral/bilateral foot or lower limb trauma/surgery, neurological issues, major cardiovascular conditions, psychiatric issues or present physical congenital or secondary abnormalities at the lower limbs were considered exclusion criteria. The participant’s characteristics are presented in Table 1.

### 2.5. Methods of Participant Preparation for Measurement and Clinician Intervention

Participant preparation for measurement and clinician intervention followed the calibration procedure. A COVID-19 safety protocol was applied both for the participants and the clinician, and after rigorous disinfection of all the device components was properly done, the participants’ feet teguments undertook the same procedure. After a complete verbal description of the device and method of data acquisition was briefly presented, written informed consent was released and signed by all participants before any assessment. Volunteers were informed about the whole procedure being noninvasive and all possible adverse reactions. Possible adverse physical reactions included local pain at strap contact, muscle pain, discomfort, cramps, fatigue, or any other psychological issues like fear or anxiety. The participants were asked to sit on a chair with the hip and knee joint flexed and the foot plantigrade on the wood board of the dynamometer plate, while the other foot was relaxed on the ground.

Based on previous research, a decrement of maximal torque was observed with a more flexed knee angle and ankle plantarflexion when an electric stimulus was applied to the gastrocnemius muscle, and greater torque with the knee at 0° of flexion [16]. Based on this electrophysiology study results on the impact of the gastrocnemius–soleus complex as the primary plantar flexor of the ankle, we decided that all measurements should take place with the knee at 90–110° of flexion to minimize its contribution and better isolate the small muscles involved in plantar flexing the ankle (flexor hallucis longus, flexor digitorum longus and the long toe flexor muscles).

The participants were asked to fully relax in order to adapt the nonelastic fiber belt that fixed the thigh to ensure that the foot remained in place and the heel did not rise up when ankle movements were performed. Another strap was placed on the dorsum of the forefoot, just above the metatarsal-phalangeal joints (MPJ), blocking the foot when analyzing the dorsiflexors strength (Figure 3, left). Placing the foot in the right position made the ankle joint the center of rotation, appreciated as a line passing through the medial and lateral malleolus and aligned with the dynamometer’s pivotal point marked on the plate (Figure 3, right). Any emotional or physical discomfort, as well as adverse reactions like pain, muscle cramps or tension during the procedure were considered and followed by cessation of the session. After complete instruction on the type of requested contraction and succession of muscle efforts, an experimental trial was initiated, without being analyzed as a valuable measurement. As the participant fully understood the procedure, the acquisition of the voltage started, and the participant received the commands to tense the muscles in order to flex the foot in the requested directions. Another trial of three MVIC was registered for both left and right ankle plantar flexors and dorsiflexors muscle groups. The clinician had in regard the duration of the acquisition which was set to 32 s and efficiently organized the succession of the three commands given to the participant. The acquisition started at least two seconds after the participant fully relaxed in order to have an initial constant offset value established. Consecutive commands followed, leaving five-second breaks between the three contraction sessions. Three consecutive maximum efforts of isometric contractions were performed for the plantar flexors group. Every isometric contraction lasted for 5 s. The same procedure was applied for the dorsiflexor muscles group. Three pedal inclinations were determined using an electronic inclinometer (0°, +5°, −5°), and three MVIC/pedal inclinations were determined during plantar flexion and dorsiflexion, resulting in twelve measurements for each participant. Fatigue was prevented by allowing a two-minute recovery break between the plantar flexors and dorsiflexors muscles’ efforts for all inclinations. The acquisition was stopped after a set time, and files were recovered from the PC for further analysis. The force cell was able to measure a positive value for force during active ankle plantar flexion and a negative value force during active ankle dorsiflexion that was transformed into voltage and further into Nm for torque.

### 2.6. Statistical Analysis

Data were presented as arithmetic mean ± standard deviation, minimum and maximum. To calculate reliability, the interclass correlation coefficient (ICC) was used. The ICC was presented together with a 95% confidence interval and *p*-value. A good reliability value was considered if ICC was over 0.72. A significant *p*-value was considered to be *p* < 0.05. Data were statistically analyzed with SPSS 25.00.

## 3. Results

### 3.1. Determination of the Pivotal Point

Once the system started, there was an offset voltage in absence of any extra load on the pedal, which was around 200 mV in our case. We used a 0.5 kg dumbbell for this procedure, since it had a shorter edge and allowed for more precise positioning.

Since the dumbbells had a hexagonal profile, the position of the dumbbell was measured using the middle point of one of the hexagonal sides. The closest point to the reference position was chosen next to a 1 cm marker (at the end of the board) in order that the dumbbell did not fall down from the pedal.

In the closest point to the reference position (upper end of the board), one could measure the largest voltage over offset value, while at the other end of the board, the measured voltage was minimal and less than the offset value. The pivotal position was the position in which the dumbbell load did not alter the offset voltage. In our case, the pivotal position proved to be at the 29 cm marker, starting from zero as the reference point (Figure 2, left).

While making the foot measurements, the line that passed through the ankle axis of rotation (represented by the medial tibial malleoli and the lateral peroneal malleoli) needed to be positioned as close as possible to the pivotal position. 

### 3.2. Calibration with Weights

The measured values obtained during calibration with weights are shown in Table A1/Appendix A containing weight, position and voltage, and the torque was calculated:(1)$$T\left[\mathrm{kgf}\cdot \mathrm{cm}\right]=m\left[\mathrm{kg}\right]\cdot \left(d_{0}-d\right)\left[\mathrm{cm}\right]$$
where *T* is the calculated torque, *m* is the dumbbell weight, *d*_0_ is the pivotal point position and *d* is the dumbbell position.

Regarding the conversion from kg·cm to the SI unit for torque N·m, the formula is:(2)$$T\left[N\cdot m\right]=T\left[\mathrm{kgf}\cdot \mathrm{cm}\right]\cdot 0.0980665$$

With torque measurements versus voltage series ordered by torque, a linear regression was performed in order to find out the proportionality constant between torque and measurement voltage, *k* [kgf⋅cm/mV]. The values should fit the equation: (3)$$T=k\cdot u-T_{0}=k\cdot \left(u-u_{0}\right)$$
where *T*_0_ is the static (remanent) torque, proportional to the offset voltage *u_0_*, and *u* is the measured voltage due to the static torque and foot torque together.

The proportionality constant was determined as *k =* 0.3422 kgf·cm/mV *=* 0.0335583563 N·m/mV = 1/29.8 N·m/mV through this procedure. While the offset needed to be constantly evaluated, since it had a little wobble due to the movements of the foot on the pedal, the proportionality constant remained reasonably stable. 

Linear regression can be performed in spreadsheet software, like Microsoft Excel [17]. or LibreOffice Calc [18], but also in computer algebra systems like MathWorks Matlab [15] or GNU Octave [14]—the second option is free software in both cases. The results of our linear regression can be seen in Figure 4.

Measured values with weights are gathered in Table A1, Appendix A containing weight, position and voltage. The torque was then calculated, and from the torque-voltage pairs, the proportionality constant was determined through linear regression. Appendix B discloses the used code of calibration, with initial values included. Our value for proportionality constant was k ≃ 1/29.8 Nm/mV.

### 3.3. Oscilloscope Software Settings, Data Collection and Processing

After calibration, the operator used the constant further on, which was the result of the calibration procedure.

The acquired voltage recorded during multiple muscle contractions was further analyzed for establishing the optimal given oscilloscope software settings that needed to be set by the operator before trials.

Since we used a particular type of oscilloscope with some particularities about data collection, our detailed settings’ protocol is fully disclosed as we used it in this configuration.

As a general guidance, one must capture between 10 and 30 s of pedal signal, depending on the intent (e.g., number of contractions/trial). In this amount of time, the volunteer performs a few contractions on demand, which will be collected as a time function. As a sample rate, a minimal 10 sps (samples/second) should be used, but a higher sample rate (1 kHz or more) is recommended. This creates the possibility of filtering out parasitic signals, coming mainly from the power network hum. If down sampling is performed after the filtering, the measurement resolution increases. This is also recommended since lower-cost oscilloscopes exhibit 8-bit resolution. As an input scale, this depends on the conditioning circuit and should be taken in accordance with the maximum expected voltage (±2 V in our case), and the coupling should be direct current (DC), only first channel used (channel A in most of the cases). We used a 10-bit setting from the oscilloscope, which provided some software smoothing of the captured voltage and raised the effective resolution.

There was no preset software sampling rate, and another particularity of this software was the collection of 32 data buffers (the last 32 waveforms from the screen) in 32 text files (as the option we used). We set the time/div parameter at 100 ms, yielding a sample frequency of 6104.5 Hz (high enough) and a record length of 32 s. For a 50 ms/div setting, the record was 16 s long (all 32 buffers), and sample frequency was double (12,209 Hz).

It is estimated that a much lower sampling frequency than the one presently used (6104.5 Hz) could be suitable as well for this kind of analysis (e.g., at least 10 Hz, but 1000 Hz is recommended in order to filter out electric network components with ease, if necessary). Our estimate of 10 Hz sampling frequency was based on the observation of acquired frequency spectrum of the torque signal. 

Summarizing, our settings were: Channel A on, DC coupling, input = ±2 V and time/div = 100 ms/div (32 s length record) or 50 ms/div (16 s length record). An example of captured voltage during one experimental MVIC with the recommended settings is described in Figure 5.

Measurements can be noisy due to the electromagnetic environment, as shown in Figure 6 (left). If noise remains after repositioning the cables, low-pass filtering with a cut-off frequency of 10–20 Hz can be performed in order to cancel the noise. After low-pass filtering with a FIR (finite impulse response filter) of order 128, having a cut-off frequency of 16 Hz, the signal shape changed as in Figure 6 (right). In a simpler manner, coiling the cables together should also help reduce the noise. Since the pedal and its load cell itself exhibited inertia, as one can see in Figure 6, aggressive low-pass filtering upon the input signal had practically no effect on the torque shape record.

The filter was designed with Matlab/Octave command b = fir1(128.,16 ∗ 2/fs), where fs is the sampling frequency.

### 3.4. Clinical Measurements and Statistical Analysis

Various angles/platform inclinations were tested using an electronic inclinometer. We defined the 0° of plate inclination as the right angle between the footplate and participant tibia (tibia being perpendicular to the dynamometer plate), +5° as 95° of ankle dorsiflexion, and −5° as 5° of ankle plantarflexion starting from reference point of 0°. There was an unstable offset voltage observed, mostly due to the participant’s inability to completely relax between contractions, or pedal mechanical remanent strains. 

We measured passive moment at rest and ankle torque at different ankle angles (0°, +5°, −5°) during three consecutive MVIC of 5 s each separated by 5 s relaxation periods (until the offset stability was gained), with the knee joint angle between 90° and 110° during plantar and dorsiflexion, creating thereafter 12 measurements for each participant, summarizing a total of 48 results. From the total of 48 results, we excluded three measurements from the dorsiflexion group and one measurement from the plantarflexion group due to reported pain at the level of strap application, resulting in 8.3% measurements loss.

Results from the three consecutive MVIC, peak torque (PT) in Nm (described as the difference between the maximum obtained level of torque and the minimum obtained level of torque) and mean (m) values (between PT resulted at the three inclinations) were registered and statistically analyzed.

When considering the measurements made on left and right ankle on both plantar flexion and dorsiflexion, all the data presented in Table 2 had good reliability, with interclass correlation coefficients (ICC) = 0.96, 95% CI (0.84;1.00), *p* < 0.0012. When separately analyzed, we found very good reliability between the left ankle and right ankle for the plantar flexion ICC = 0.89, 95% CI (0.52;0.99), *p* = 0.002, and between left ankle and right ankle mean for the dorsiflexion ICC = 0.94, 95% CI (0.67;1.00), *p* = 0.001. We found good reliability between dorsiflexion and plantarflexion for the left ankle ICC = 0.96, 95% CI (0.82;1.00), *p* < 0.001, and good reliability between dorsiflexion and plantarflexion in the right ankle ICC = 0.84, 95% CI (0.21;0.99), *p* = 0.013.

When all measurements at 0° were analyzed, we found very good reliability ICC = 0.92, 95% CI (0.49;1.00), *p* = 0.005, as well as for all −5° measurements ICC = 0.87, 95% CI (0.12;0.99), *p* = 0.019. Despite the relatively high ICC value for all +5° measurements ICC = 0.73, 95% CI (−0.80;0.98), *p* = 0.082, statistical significance was not reached.

Based on comparison of the results as shown in Table 2, Participant 1 (left foot) was the most reliable, and we used it as a reference measurement (Figure 7, left). An example of captured error due to reported pain during testing is represented as a graph in Figure 7 (right).

Both human errors (participant and/or operator) and apparatus errors were simple to detect in time graphs (oscillograms) captured with the given oscilloscope software, so it was possible to establish if the measurement was properly done. This would not have been possible with simpler means (e.g., a simple voltmeter instead of an oscilloscope).

Operator eye-detected errors due to participants were: change of position between contractions (foot on pedal or trunk forward/backward sway), delayed reactions to the vocal commands and/or static tremor. 

Errors due to operator were detected by graph inspection by a second operator, mainly unequal temporization of vocal commands in relation with agreed time markers.

Detected errors due to the apparatus were a delay of 2–3 s between start command pressed by operator and effective acquisition start, electric network noise later corrected by filtering, remanent strains in pedal, hysteresis in pedal characteristics and force variations obtained between initial position and various pedal inclination angles.

## 4. Discussion

Our results showed that in absence of particular training of the testator, this device was an objective method for measuring ankle torque during MVIC, which was more reliable compared to manual muscle testing based on comparison of ICC values with previous studies [2]. 

When analyzing the literature, we found few studies describing calibration models when using portable electronic custom-made dynamometers for measuring lower limb strength/torque. Calibration requires a rigorous procedure and should precede any clinical assessment, including configuration of the software settings. 

When assessing Reeves’ work [7], we observed two issues that needed in-depth research on our available replica of the portable device provided by the manufacturer. 

The first of them was the presence of an offset voltage which distorted the results. The calibration and measurement without taking into consideration the initial offset voltage would have led to relatively large errors, so we made use of initial offset cancelling. The second issue was the position of the pivotal point. On our custom-made apparatus, the pivotal line was supposed to be the projection of the pedal axis onto the pedal surface, as provided by the manufacturer. Our measurement with the offset method concluded that the pivotal point position was slightly different (1–2 cm) from the mechanical pivotal point initially marked on the device, which was of importance since it affected the external moment arm distance and therefore the calculation of joint torque. This was even more important when considering smaller feet sizes. 

Marsh et al. used weights on the plate for calibration, and by multiplying each weight with the brace of force, torque was then calculated in the same manner. No disclosure was done regarding the direct current measurement apparatus used, although other devices for evaluating electromyographic (EMG) activity were indicated [6]. There is no indication on time recording of ankle torque in the mentioned study. We used voluntary contractions and not electrically stimulated contractions, and measurements were made electronically with a strain gauge and an oscilloscope. The offset level (direct current) of the recording system was taken into consideration to cancel passive torque as previously done by Marsh. In the same study, sensor calibration was obtained with weights at various positions on the pedal, but we performed linear regression with all weight measurements in order to obtain the most accurate proportionality constant value. We obtained the time recordings, which allowed us further analysis. An advantage of our approach is that the series of isometric contractions were available as time functions, thus opening the door for more in-depth analysis on torque variation over time. A decrease in torque could be seen on the raw torque-time traces after peak torque had occurred.

Establishing the precise initial offset of the device, we managed to avoid misinterpretation of the acquired data, enabling more reliable measurements. The linear regression showed that the calibration technique was consistent. The portable electronic custom-built dynamometer used in this study was found to be a reliable and effective device for measuring ankle joint torque when precise calibration was performed prior to trials. 

With the reported values, a series of measurements were performed, with various settings of electronic equipment. We found an optimal setting that led to a 32 s record of voltage, which was further processed. We worked for testing in a 5 s scenario (pause–contraction–pause), obtaining a number of three contractions in this time interval. 

There are other measurement scenarios previously used in other studies, with longer pauses between contractions [8] and similar single contraction time [19].

The portability of both hardware components and software for the used dynamometer could provide efficiency in both research and clinical use. The described apparatus, being a portable device, could open access for on-site ankle torque measurements, making even homebound or immobile participants available for testing.

Data similar to that of the present study were obtained in a French study where ankle torque was measured using handheld dynamometry during plantar flexion [5]. 

When compared to an isokinetic apparatus able to measure muscle work at different angles and speed, our device could be considered limited (it only determined muscle isometric strength defined as peak torque during a single selected angle). When compared to isokinetic testing, our device is lighter and smaller, making it hand-transportable (7.5 kg). No specific data was found on weight and dimensions of other similar devices, other than the one used by Marsh (approximately 3 kg). 

Similar data were obtained when ankle torque (in Nm) was measured in humans using similar custom-made dynamometers [8], but we need to consider that different devices, despite being similar in construction, could give different results when measuring ankle torque due to force being changeable in time even in the same subject (participant) and due to the electronic means of the measurement system (load cell, oscilloscope, software parameters, etc.)

Despite the technical limitations, devices similar to our apparatus have been used in previous studies in a more simple or complicated fashion. When analyzing the literature, we found few studies describing calibration models when using portable electronic custom-made dynamometers for measuring lower limb strength/torque. Calibration requires a rigorous procedure and should precede any clinical assessment, including configuration of the software settings. The aim of our study was to calibrate a given manufactured dynamometer (replica of previous device used by Reeves et al.) and have its reliability tested, as we did not find in other papers using similar devices any data on the validity/reliability of this type of apparatus when measuring ankle torque. The other aim was to release a measurement protocol with the given custom-made apparatus (including not just the calibration procedure, but also a patient preparation procedure, type of muscle efforts, the resulted time graphs after mathematical data processing and examples of encountered errors during measurements that can alter the validity of the results).

This work will contribute to designing models for ankle torque measurement and foot function assessment in humans. New foot strength normative data for healthy individuals could further be established in various populations using our model. 

Future work should be considered for statistical analysis on a statistical representative group. In our case, reliability for the +5 measurements could not be evaluated, possibly due to the small sample.

## 5. Conclusions

Portable electronic dynamometry is a reliable tool for measuring ankle torque in clinical practice, bringing higher accuracy in diagnosis, treatment monitorization and assessment of the treatment outcomes. Such devices could help develop a normative database related to healthy populations for foot and ankle status parameters. As it involves lower costs when compared to those of isokinetic testing, portable dynamometry, due to its maneuverability, could open the possibilities for on-site testing. 

## Figures and Tables

**Figure 1 sensors-22-00135-f001:**
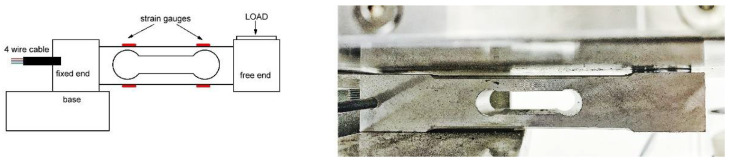
A load cell representation (**left**) and the load cell used in our device (**right**).

**Figure 2 sensors-22-00135-f002:**
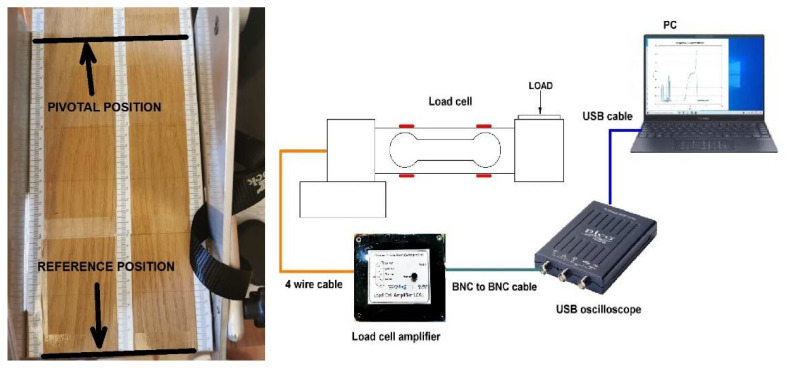
Pedal with distance ruler (**left**), block diagram of the measurement system (**right**).

**Figure 3 sensors-22-00135-f003:**
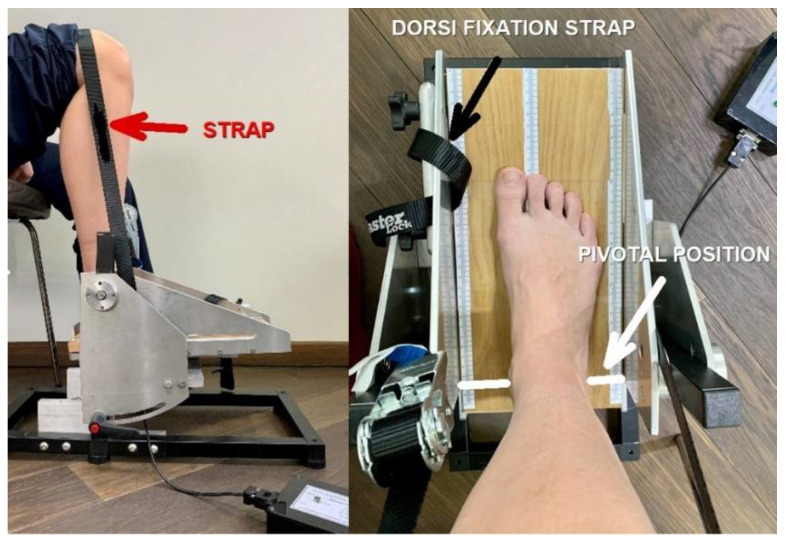
The participant’s position on the chair, fixation of straps (**left**) and foot position on the plate, with ankle malleoli axis above the pivotal line mark (**right**).

**Figure 4 sensors-22-00135-f004:**
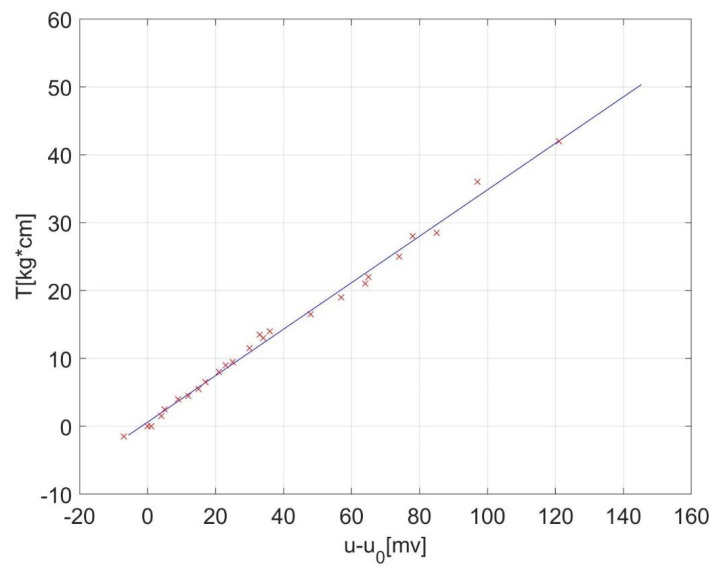
The result of linear regression (line) versus calibration measurements (crosses).

**Figure 5 sensors-22-00135-f005:**
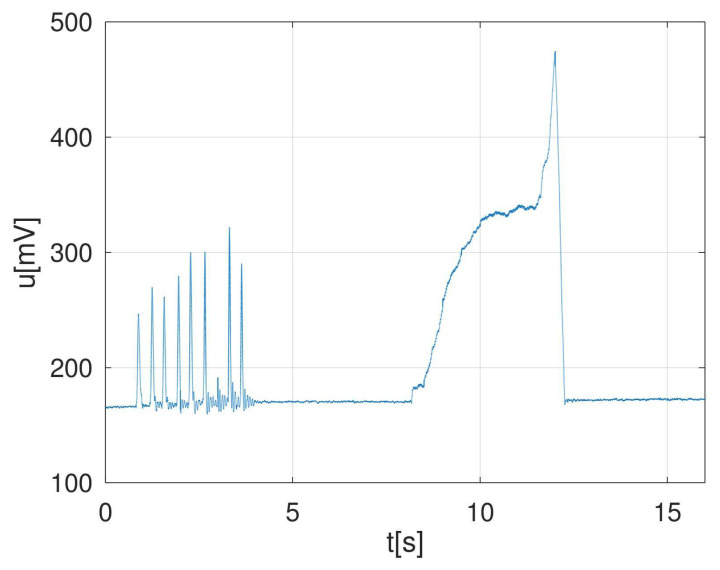
An example of captured voltage during multiple contractions followed by a single MVIC.

**Figure 6 sensors-22-00135-f006:**
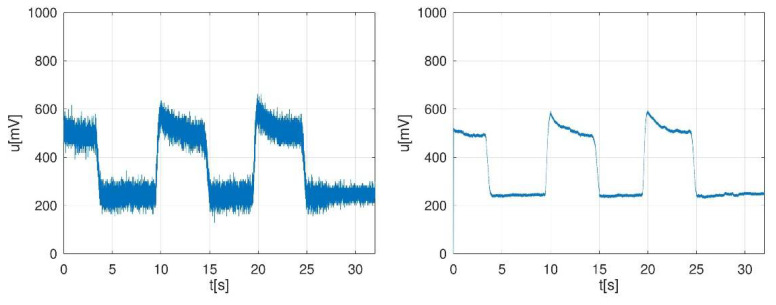
An example of noisy captured signals (**left**), captured signals after filtering (**right**).

**Figure 7 sensors-22-00135-f007:**
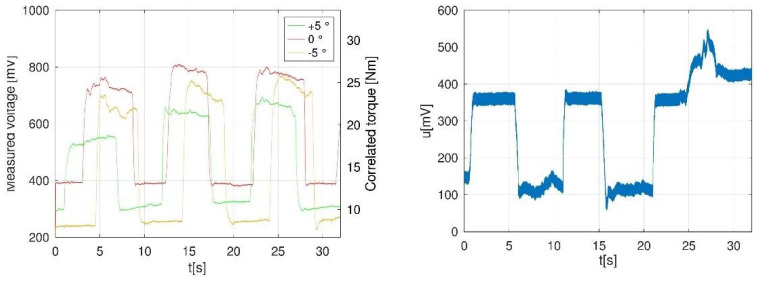
Three captured time series (5 s of contraction and 5 s of relaxation between contractions) for the same participant, right foot, with various initial pedal inclination (+5°, 0°, −5°) during plantar flexion, and resulted voltage in mV and correlated torque in Nm (**left**); example of error of measurement due to participant reporting pain which further resulted in improper discipline (**right**).

**Table 1 sensors-22-00135-t001:** The participants demographic and anthropometric characteristics ^1^.

Parameters	Mean ± Standard Deviation	Minimum	Maximum
**Age (years)**	39.2 ± 15.1	21	58
**Height (m)**	1.72 ± 0.14	1.60	1.92
**Weight (kg)**	76. ± 32.2	57	124
**Foot length (cm)**	24.5 ± 2.8	-	-

^1^ Group size *n* = 4; male, no. (%): 1 (25).

**Table 2 sensors-22-00135-t002:** Individual participant data showing peak torque (Nm) during MIVC for both plantarflexion and dorsiflexion for four participants at 0°, +5° and −5°.

	Peak Torque (Nm)	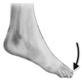	Peak Torque (Nm)	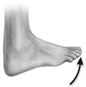
Plantarflexion	Dorsiflexion
	Left Ankle	Right Ankle	Left Ankle	Right Ankle
Subject	0°	5°	−5°	m	0°	5°	−5°	m	0°	5°	−5°	m	0°	5°	−5°	m
1	27.1	25.5	22.2	24.9	X	26	23.3	24.6	17.7	16.8	20.9	18.4	28.8	11.6	17.8	19.4
2	22	24.1	19.4	21.8	21.3	22.9	17.9	20.7	14.1	12.1	14.1	13.4	12	10.3	X	11.5
3	42.9	43.7	52	46.2	44.8	39.3	31	38.3	18.3	X	24.5	21.4	21.2	16.7	18.7	18.8
4	23.9	15	17.7	18.8	29.6	32.9	31.1	31.2	29.2	24.4	20.7	24.7	27.4	14.6	X	21

Note: m—arithmetic mean, X—captured errors.

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
