# Peer review of "A Custom-Made Lower Limb Dynamometer for Assessing Ankle Joint Torque in Humans: Calibration and Measurement Procedures"

_sensors, 2021, doi:10.3390/s22010135_

Round 1

Reviewer 1 Report

This paper presents a custom-made dynamometer for human ankle joint torque assesment. The dynamometer has been designed and tested by 4 volunteers. The calibration and measurement procedure is introduced. Here are some issues to be considered before consideration of publication:  1. In the 1st section Introduction, the state of the art of ankle torque measurement devices have not been summarized well. I can not find out what principles other researchers used to design the devices. Meanwhile, the authors should clearly point out their contribution or innovation at the end of this section. The objectives of this study do not equal to the contribution or innovation. 2. Some figures in this manuscript are of low quality, such as Figure 1 right, Figure 5 and Figure 7. Most figures are not clear enough, and the axis information are too small. It's hard to get information from the figures. 3. To show the advantages of this device, the authors should compare the results to the results measured from some other devices. In the results part, there are only results of 4 participants from the designed device. 4. The authors emphasized that this is a portable device. But I do not think it is a portable one since it is not lightweight enough and is not wearable.

Author Response

Thank you for carefully reviewing our manuscript.

We hope that we succeeded to match every request.

FP 

Reviewer 2 Report

The purpose of this study is to Establishing the calibration procedure for the existing portable custom-made electronic dynamometer and standardization of a method of measurement of strength around ankle joint. Another objective was to determine ankle torque during maximal voluntary isometric contractions (MVIC) for both plantar-flexors and dorsiflexors muscle groups, and establish reliability for this portable dynamometer.

The work is interesting, but some improvements may be done:

The introduction is not easy to read. There are sentences that are not connected to the rest of the text. It would be interesting to reflect about the importance of measuring the ankle torque in rehabilitation practice.

The methods are described in a way that seems the authors are doing a literature review. It would be more interesting if the authors explained their choices, instead of saying that “there are other options available”. Again in the results, the author say “the linear regression can be performed in different software, like…” since they are not explaining their choices, this does not make any sense.

The quality of the figures, the font size of the axis are not well to read.

Table 2 is hard to understand. The data do not present units, there is something that looks like an average but without standard deviation. The picture of the foot is not well positioned, and this creates some confusion to understand the table. The “X” in it has no explanation.

The discussion is confused. The information is there but does not follow a logic sequence. Some sentences are lost in the middle.

Author Response

(The authors gave the same response as above.)

Round 2

Reviewer 1 Report

This paper presents design and validation of a portable electronic dynamometry as a reliable tool for measuring ankle torque in clinical practice. I am pleased with the responses and the revised manuscript.